# Unlearning Shapley: Data Valuation through Machine Unlearning

## Abstract

Data valuation is essential for understanding and improving machine learning models. However, existing approaches such as Shapley-value-based retraining or influence functions are either computationally prohibitive or require full access to the training data, which is often unrealistic in practice. This challenge is particularly pressing in real-world settings such as data markets, federated environments, or compliance with the right to be forgotten, where only partial access to data subsets is available. We introduce Unlearning Shapley, a framework that adapts machine unlearning for both full-data and partial-data valuation. Instead of repeatedly retraining on all subsets, our method leverages a single pre-trained model and applies approximate unlearning to remove the effect of the target data, thereby estimating its marginal contribution. This design uniquely enables valuation when the rest of the training data is inaccessible, offering a privacy-compliant and practically deployable solution. Through theoretical analysis, we show the connection between Unlearning Shapley and classical Shapley values, and we provide bias and error bounds for our estimator. Experiments on benchmark vision datasets and large-scale language models demonstrate that Unlearning Shapley achieves comparable or superior performance to state-of-the-art methods in identifying influential or noisy data, while reliably extending to the partial-data setting where existing approaches fail. Our study highlights the importance of partial data valuation and extends the applicability of machine unlearning beyond privacy to equitable and transparent data markets.

## 1 Introduction

Data valuation quantifies the contribution of individual data points to model performance and is fundamental to building transparent and economically sustainable machine learning systems. As data markets mature and privacy regulations such as the GDPR establish the *right to be forgotten*, reliable data valuation becomes increasingly important for fair compensation, model debugging, and regulatory compliance Azcoitia et al. (2023); Xia et al. (2023); Jia et al. (2019a).

Despite significant progress, existing valuation methods face two fundamental limitations. First, they typically require *retraining on many subsets of the data*. Shapley-value-based approaches Shapley et al. (1953); Ghorbani & Zou (2019) provide axiomatic fairness guarantees, but computing them exactly involves training exponentially many models. Approximation techniques based on Monte Carlo sampling Ghorbani & Zou (2019); Kwon & Zou (2021); Jia et al. (2019a) or influence functions Koh & Liang (2017); Pruthi et al. (2020) reduce this burden, yet remain computationally heavy and often impractical for modern large-scale models.

Existing methods assume complete access to the training data. In practice, this assumption often fails. Three common scenarios highlight the problem. First, in the data market, vendors wish to evaluate their own contributions without accessing competitors' data. For federated learning, institutions need to assess local data utility without sharing sensitive datasets. Additionally, in privacy-preserving machine learning, organizations must value data scheduled for deletion without retaining it. These scenarios form the broader class of partial data valuation, which, despite its prevalence, has received little direct methodological support.

We introduce Unlearning Shapley, a framework that replaces retraining with machine unlearning to enable both full and partial data valuation. The key idea is to approximate the utility of training on

subset $S$ not by retraining from scratch, but by starting from a pre-trained model on $N$ and *unlearning* the complement set $N \setminus S$. This reformulation allows valuation using only the pre-trained model and the target subset, making it applicable in privacy-sensitive and federated settings where complete data access is infeasible.

From a theoretical perspective, we prove that Unlearning Shapley is equivalent to the classical Shapley value under mild assumptions, and we provide error bounds that disentangle approximation bias from Monte Carlo sampling variance. Practically, we design two algorithms: one for full-data settings with efficient Monte Carlo estimation, and one for partial-data settings requiring only the target subset. Our experiments on image classification benchmarks and large-scale language models demonstrate that Unlearning Shapley reliably identifies influential and noisy data, and uniquely succeeds in partial-data scenarios where existing methods fail.

Our main contributions are:

- We identify partial data valuation as a critical but overlooked challenge, and provide the first practical solution via machine unlearning.
- We establish the theoretical equivalence between Unlearning Shapley and classical Shapley values, together with error bounds that clarify the sources of approximation error.
- We develop algorithms for both full-data and partial-data settings, enabling data valuation without requiring access to the entire training dataset.
- We empirically validate Unlearning Shapley across vision and language tasks, showing its effectiveness in noisy data detection and its unique applicability in privacy-preserving scenarios.

## 2 RELATED WORK

### 2.1 DATA VALUATION METHODS

Data valuation has emerged as a key research area for quantifying the contribution of individual data points to machine learning (ML) models. Among existing approaches, two prominent families are Shapley-value-based methods and influence-function-based methods.

Shapley-value-based methods, inspired by cooperative game theory, provide a principled and axiomatic foundation for fair credit assignment Jia et al. (2019b). However, the exact computation of Shapley values requires exponential time in the dataset size, making approximation indispensable Mitchell et al. (2022). Data Shapley Ghorbani & Zou (2019); Jia et al. (2019a) introduced Monte Carlo sampling over random permutations to estimate marginal contributions, which has since become a standard approach. Variants have been proposed to improve efficiency and interpretability. Beta Shapley Kwon & Zou (2021) relaxes the efficiency axiom to reduce variance and provide optimal importance weights. KNN Shapley Jia et al. (2019a) exploits the structure of the $k$-nearest neighbor classifier to compute exact Shapley values in $O(n \log n)$ time, enabling large-scale applications.

More recently, Threshold KNN Shapley Wang et al. (2023) extends this idea to privacy-preserving scenarios and introduces differential privacy guarantees with minimal utility loss. Other extensions, such as asymmetric Shapley Zheng et al. (2024), probability-based Shapley (P-Shapley) Xia et al. (2024), and weighted Shapley (FW-Shapley) Panda et al. (2024), adapt the framework to account for structural dependencies, probabilistic outputs, and reweighted coalition sampling, respectively. Despite these advances, concerns remain about when Shapley values truly reflect useful data quality. Recent work Wang et al. (2024a) shows that Data Shapley can fail to outperform random selection in data subset tasks unless specific conditions on utility functions hold, highlighting the importance of theoretical understanding of its limitations.

Another type of data valuation methods are Influence-function-based methods. These methods provide an alternative rooted in robust statistics Koh & Liang (2017). They measure the effect of perturbing a training example on model parameters or test predictions. TracIn Pruthi et al. (2020) traces training trajectories to efficiently approximate influence using saved checkpoints and gradient information. Simplified methods Pruthi et al. (2020); Jiao et al. (2024) drop the Hessian inversion term, though scalability remains challenging for modern large-scale models. LOGIX Choe et al. (2024) incorporates low-rank adaptation (LoRA) to approximate influence on compressed subspaces,

substantially reducing computation costs. Reinforcement-learning-based selection Yoon et al. (2020) has also been proposed, but training the selector itself can be computationally prohibitive. Overall, while influence-based approaches provide valuable interpretability, their dependence on gradient and Hessian computations limits their practicality for foundation-scale models.

In light of these challenges, Unlearning Shapley builds on Shapley-value-based approaches while avoiding retraining and large-scale gradient computations. Instead, it leverages machine unlearning to estimate marginal contributions, combining the axiomatic fairness of Shapley values with the efficiency of subtractive updates.

## 2.2 MACHINE UNLEARNING

Machine unlearning was originally motivated by privacy regulations such as GDPR's "right to be forgotten," enabling trained models to remove the influence of specific data while retaining performance on remaining data Wang et al. (2024b); Xu et al. (2024). Solutions can be categorized into exact and approximate approaches Wang et al. (2024b); Xu et al. (2024). Exact unlearning often involves retraining without the deleted data, sometimes accelerated through distributed or federated learning Halimi et al. (2022); Li et al. (2024). Approximate unlearning modifies trained models directly, typically through parameter updates. For instance, gradient ascent on the forgotten data combined with regularization on retained data has been proposed Kurmanji et al. (2023); Triantafillou et al. (2024). To mitigate dependence on access to the retained dataset, divergence-based techniques replace retraining with constraints such as Kullback–Leibler divergence between original and unlearned models Zhang et al. (2024).

Recent works have extended unlearning to large models. Studies on LLM unlearning Yao et al. (2023); Maini et al. (2024); Shi et al. (2024) demonstrate that targeted unlearning can be achieved with surprisingly few update steps, while maintaining general capability. Xu et al. (2024) highlights challenges such as balancing utility preservation, scalability, and provable guarantees. These advances establish unlearning as not only a tool for privacy and security but also a means for model adaptability in dynamic environments.

Our work connects machine unlearning with data valuation. By replacing costly retraining or gradient-based influence estimation with efficient unlearning procedures, Unlearning Shapley provides a scalable and principled framework for attributing data value in large-scale ML models.

## 3 METHODS

### 3.1 THE SHAPLEY VALUE

Given a dataset $N := \{d_i\}_{i=1}^n$ of $n$ data points, the Shapley value quantifies the contribution of each point $i$ by averaging its marginal utility across all possible subsets:

$$\phi_i = \sum_{S \subseteq N \setminus \{i\}} \frac{|S|!(n - |S| - 1)!}{n!} \big(v(S \cup \{i\}) - v(S)\big) \tag{1}$$

$$= \frac{1}{n} \sum_{S \subseteq N \setminus \{i\}} \binom{n-1}{|S|}^{-1} \big(v(S \cup \{i\}) - v(S)\big),$$

where $v(S)$ denotes the utility of subset $S$. Following standard practice, we define $v(S) := \mathbb{E}[\text{Perf}(\mathcal{M}_S, \mathcal{T})]$ as the expected performance of a model $\mathcal{M}_S$ trained from scratch on $S$ and evaluated on a fixed test set $\mathcal{T}$. For simplification, if the model is not evaluated on a specific dataset, we use $\text{Perf}(\mathcal{M})$, ignoring $\mathcal{T}$. For the metrics, we use accuracy for classification and negative perplexity $\text{Perf}(.)$ for text generation tasks because a higher output of the utility function represents a higher marginal contribution. This expectation absorbs training randomness.

The Shapley value can be equivalently expressed via permutations:

$$\phi_i = \frac{1}{n!} \sum_{\pi \in \Pi} \big[v(S_\pi^i \cup \{i\}) - v(S_\pi^i)\big], \quad S_\pi^i := \{j \in N : \pi(j) < \pi(i)\}, \tag{2}$$

---

**Algorithm 1** Unlearning Shapley for **Full Data Valuation**

---

**Input:** Dataset $N = \{d_1, \ldots, d_n\}$, pre-trained model $\mathcal{M}_{\text{full}}$, test set $\mathcal{T}$, MC samples $M$.
**Output:** Shapley values $\{\phi_i\}_{i=1}^n$
1: Define $u(S) := \text{Perf}(\widehat{\mathcal{M}}_S)$, where $\widehat{\mathcal{M}}_S$ is $\mathcal{M}_{\text{full}}$ after unlearning $S$
2: Initialize $\phi_i \leftarrow 0$ for all $i$
3: **for** $m = 1$ **to** $M$ **do**
4:     Sample a permutation $\pi$ of $\{1, \ldots, n\}$; set $S \leftarrow \emptyset$
5:     **for** $k = 1$ **to** $n$ **do**
6:         $i \leftarrow \pi(k)$;    $\Delta \leftarrow u(S) - u(S \cup \{i\})$                          *// marginal drop*
7:         $\phi_i \leftarrow \phi_i + \Delta$;    $S \leftarrow S \cup \{i\}$
8:     **end for**
9: **end for**
10: **Return** $\{\phi_i/M\}_{i=1}^n$

---

where $S_\pi^i$ represents the set of points preceding $i$ in permutation $\pi$. While Monte Carlo (MC) methods can estimate these values by sampling permutations, the computational cost of repeated model retraining remains prohibitive for practical applications.

### 3.2 UNLEARNING SHAPLEY

Unlearning Shapley transforms the expensive retraining paradigm into an efficient unlearning-based approach. Our key insight is that instead of training a model on a subset $S$ from scratch, we can approximate its performance by starting with a pre-trained model $\mathcal{M}_{\text{full}}$ and unlearning the complement set $N \setminus S$. Formally, we assume:

$$v(S) = \text{Perf}(\mathcal{M}_S) \approx \text{Perf}(\widehat{\mathcal{M}}_{N \setminus S}) \tag{3}$$

where $\widehat{\mathcal{M}}_{N \setminus S}$ denotes the model obtained by unlearning the complementary set from $\mathcal{M}_{\text{full}}$. This approximation holds when unlearning effectively removes data influence while preserving performance on retained data.

**Theoretical Analysis.** To formalize our approach, we define the unlearning utility function $u(S) := \text{Perf}(\widehat{\mathcal{M}}_S)$, which measures performance after unlearning the subset $S$. The key theoretical result is that the traditional Shapley value can be reformulated using unlearning operations. Starting from the marginal contribution $v(S_\pi^i \cup \{i\}) - v(S_\pi^i)$ and applying our approximation equation 3, we obtain $u(T_\pi^i) - u(T_\pi^i \cup \{i\})$ where $T_\pi^i = N \setminus (S_\pi^i \cup \{i\})$ are the successors of $i$.

Through a bijection argument over permutations—where reversing any permutation maps successors to predecessors—we derive the Unlearning Shapley formula:

$$\phi_i = \mathbb{E}_{\pi \in \Pi}[u(S_\pi^i) - u(S_\pi^i \cup \{i\})] \tag{4}$$

This expression has an intuitive interpretation: the value of data point $i$ equals the expected performance drop when $i$ is additionally unlearned. Valuable data points cause significant performance drops, yielding high positive Shapley values.

**Approximation Quality.** Under mild assumptions (detailed in Appendix A), the deviation between our Unlearning Shapley and the exact retraining-based Shapley value is bounded by:

$$|\hat{\phi}_i - \phi_i| \leq 2\varepsilon_u + B_{\text{rand}} + O(1/\sqrt{M}) \tag{5}$$

where $\varepsilon_u$ captures the unlearning approximation error, $B_{\text{rand}}$ accounts for training randomness, and $M$ is the number of Monte Carlo samples. The last term vanishes as $M$ grows, while the first two are inherent to the problem.

For application, we implement two valuation modes addressing different real-world scenarios:

**Full Data Valuation** (Algorithm 1): When complete dataset access is available, we employ Monte Carlo sampling over permutations. For each permutation, we sequentially unlearn data points and measure marginal performance drops. This approach scales efficiently since unlearning is typically much faster than retraining.

---

**Algorithm 2** Unlearning Shapley for **Partial Data Valuation**

---

**Input:** Target dataset $\mathcal{D}_{\text{tgt}}$, pretrained $\mathcal{M}_{\text{full}}$ on $N = \{\mathcal{D}_{tgt}, \mathcal{D}_{retain}\}$, test set $\mathcal{T}$
**Output:** Shapley value $\phi_{\text{tgt}}$ of $\mathcal{D}_{\text{tgt}}$
 1: Obtain $\widehat{\mathcal{M}}_{\text{tgt}} \leftarrow$ unlearn $\mathcal{D}_{\text{tgt}}$ from $\mathcal{M}_{\text{full}}$
 2: Initialize a random model $\mathcal{M}_{\text{rand}}$                                *// proxy for $u(N)$*
 3: Train from scratch on $\mathcal{D}_{\text{tgt}}$ to get $\mathcal{M}_{\text{tgt}}$                *// proxy for $\widehat{\mathcal{M}}_{retain}$*
 4: $u(\mathcal{D}_{\text{tgt}}) \leftarrow \text{Perf}(\widehat{\mathcal{M}}_{\text{tgt}})$
 5: $u(\mathcal{D}_{\text{retain}}) \leftarrow \text{Perf}(\mathcal{M}_{\text{tgt}})$
 6: $u(N) \leftarrow \text{Perf}(\mathcal{M}_{\text{rand}}); u(\emptyset) \leftarrow \text{Perf}(\mathcal{M}_{\text{full}})$
 7: $\phi_{\text{tgt}} \leftarrow \frac{1}{2}\big[u(\emptyset) - u(\mathcal{D}_{\text{tgt}}) + u(\mathcal{D}_{\text{retain}}) - u(N)\big]$
 8: **return** $\phi_{\text{tgt}}$

---

**Partial Data Valuation** (Algorithm 2): This novel capability addresses scenarios where only the target dataset $\mathcal{D}_{\text{tgt}}$ and pre-trained model are accessible, which is common in data markets, federated settings, and privacy-compliant applications. Previous methods fail here as they require full dataset access for retraining or gradient computation.

We reformulate valuation as a two-player cooperative game between $\mathcal{D}_{\text{tgt}}$ and $\mathcal{D}_{\text{retain}} = \mathcal{D}_{\text{full}} \setminus \mathcal{D}_{\text{tgt}}$, yielding:

$$\phi_{\text{tgt}} = \frac{1}{2}\big[v(N) - v(\mathcal{D}_{\text{retain}}) + v(\mathcal{D}_{\text{tgt}}) - v(\emptyset)\big]$$
$$\simeq \frac{1}{2}\big[u(\emptyset) - u(\mathcal{D}_{\text{tgt}}) + u(\mathcal{D}_{\text{retain}}) - u(N)\big]$$

We approximate these utilities without accessing $\mathcal{D}_{\text{retain}}$: $u(\emptyset)$ uses the full model, $u(\mathcal{D}_{\text{tgt}})$ via unlearning, $u(\mathcal{D}_{\text{retain}}) \approx \text{Perf}(\mathcal{M}_{\text{tgt}})$ where $\mathcal{M}_{\text{tgt}}$ trains only on $\mathcal{D}_{\text{tgt}}$, and $u(N)$ uses random initialization.

**Approximate Unlearning.** Since exact unlearning typically requires full dataset access, we employ approximate unlearning via gradient ascent with stability regularization:

$$\mathcal{L}_{\text{unlearn}} = \underbrace{-\mathcal{L}_{\text{CE}}(\mathcal{D}_{\text{unlearn}})}_{\text{Gradient ascent}} + \lambda_1 \underbrace{\|\theta - \theta_{\text{full}}\|^2}_{\text{Parameter stability}} + \lambda_2 \underbrace{D_{\text{KL}}(f_\theta(x) \| f_{\theta_{\text{full}}}(x))}_{\text{Output consistency}}, \quad x \sim \mathcal{T}$$

where the first term reverses learning on target data, while regularizers preserve model utility. Crucially, we use test set $\mathcal{T}$ for output consistency rather than $\mathcal{D}_{\text{retain}}$, enabling partial valuation without full data access. Notably, we use reversed cross-entropy loss for gradient ascent in this paper.

## 4   EXPERIMENT ON DATA VALUATION

### 4.1   DATASETS AND MODELS

We adopted five benchmark datasets and a combined large-scale text dataset to evaluate the generalizability and practicality of Unlearning Shapley. (1) FashionMNIST (FMNIST)Xiao et al. (2017): contains 60,000 training examples and 10,000 testing examples. Each example is a 28 x 28 grayscale image with a label from ten classes. (2) CIFAR-10: comprises 60,000 color images with a size of 32 x 32 in 10 classes. (3)AdultBecker & Kohavi (1996): contains 45,225 data examples with 16 attributes, with the task of predicting whether the individual incomes exceed 50k. (4) Breast CancerMangasarian & Wolberg (1990): includes 569 data examples with 30 features for predicting breast cancer. (5)MNISTDeng (2012): comprises handwritten digits with a training set of 60,000 examples, and a test set of 10,000 examples.

For MNIST, Adult, and Breast Cancer, we randomly select 500 data examples for evaluation and 100 data examples for testing. For CIFAR-10 and FashionMNIST, we use 1,000 data examples for training and the other 100 examples for testing.

Besides the benchmark datasets, we construct a large-scale text dataset, Text Corpus, for partial data valuation. Text Corpus is a combination of data examples from CosmopediaBen Allal et al. (2024),

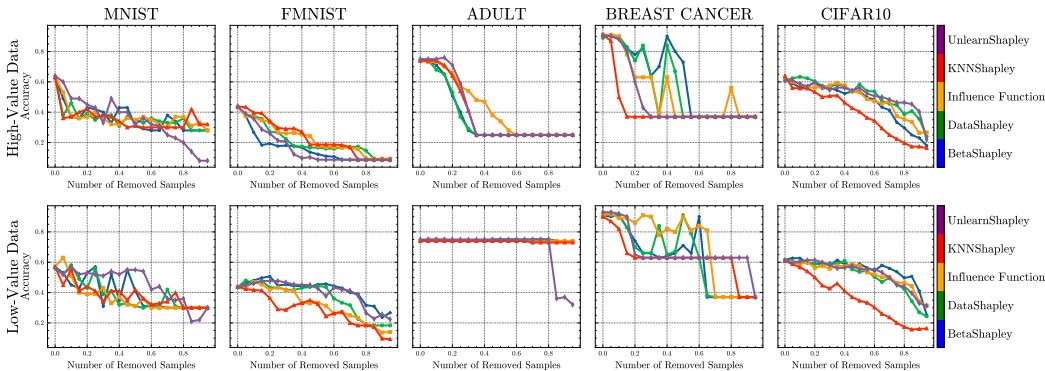

Figure 1: **Data Removal**. For data removal tasks, we increasingly remove data with the most or least values obtained by various methods and retrain a model on the remaining data to compute the metrics on the test set.

WikitextMerity et al. (2016), and Fineweb. Wikitext contains text from websites. Cosmopedia is a synthesized text corpus of textbooks, blog posts, stories, and posts. We collected 1,000,000 data points from each dataset and merged the 3,000,000 examples. Additionally, we use TruthfulQA Lin et al. (2022) as the test set.

For Unlearning Shapley implementation, we trained task-specific models: MLPs for FashionMNIST, Adult, and Breast Cancer; ResNet-18 for CIFAR-10; LeNet for MNIST; and GPT2-XL (1.5B) for the Text Corpus. Detailed training parameters are provided in Appendix B.

## 4.2 DATA REMOVAL EXPERIMENT

The data removal experiment serves as a fundamental test of whether a valuation method correctly identifies the most influential training examples. We progressively remove data points based on their computed values and observe the resulting model degradation, where an effective valuation method should induce a rapid accuracy decline when removing high-value points and a minimal impact when removing low-value points.

We evaluate the quality of a valuation ranking by its effect on retraining after removal. For each method, we sort training points by their estimated value and progressively remove either the top-ranked ("high-value") or bottom-ranked ("low-value") fraction, retraining at each step and reporting test accuracy. A good ranker induces a sharp accuracy drop when removing top-valued points and a flat curve when removing bottom-valued points. We compare Unlearning Shapley to KNNShapley, Influence Function, DataShapley, and BetaShapley, detailed as follows:

- **Data Shapley**Ghorbani & Zou (2019): Data Shapley introduces Monte-Carlo sampling to sample data permutations for the retraining until the Shapley values are converged.
- **Beta Shapley**Kwon & Zou (2021): Beta Shapley generalizes Data Shapley by relaxing the efficiency axiom, offering reduced noise and optimal importance weights for subsampling.
- **Influence Function**Feldman & Zhang (2020): We drop the Hessian matrix due to the computing complexity, and the final IF score is $IF(x_{train}, x_{test}) = \nabla L(\theta, x_{train}) \cdot \nabla L(\theta, x_{test})$.
- **KNN Shapley**Jia et al. (2019a): KNN Shapley, originally designed for the KNN classifier, proposes a KNN utility that computes the distance between the training data examples and testing examples, avoiding the multiple re-training process.

For the data removal tasks, we computed and sorted the data values in descending order. Subsequently, data with the highest or lowest values were systematically eliminated, ranging from 0% to 95% of the entire data set.

The high-value removal results in Figure 1 top row reveal compelling patterns in how different methods concentrate value attribution. On Breast Cancer, Unlearning Shapley demonstrates one of

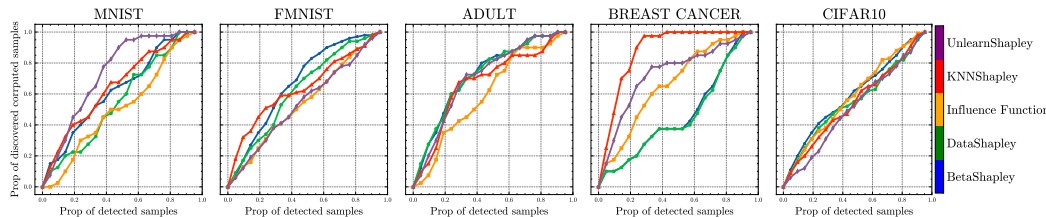

Figure 2: **Corrupted Data Detection**. For corrupted data detection tasks, we randomly flip the label of 20% of data points as noisy data. The data are split into bins of 5% of the whole data. We compute the data value using different methods and sort these values in descending order.

the steepest initial declines, with accuracy dropping from 82% to approximately 65% after removing just 20% of top-valued points. This sharp degradation indicates that Unlearning Shapley successfully identifies a concentrated set of highly influential examples. The method's performance is particularly notable when compared to Data Shapley, Beta Shapley, and Influence Function, which exhibit more gradual declines (reaching only nearly 70% at the same removal threshold), suggesting these methods distribute importance more uniformly across the dataset.

The superiority of Unlearning Shapley becomes even more pronounced on MNIST, where the method maintains the steepest sustained decline throughout the entire removal range. On MNIST, while most methods converge to similar endpoints after 95% removal, Unlearning Shapley's trajectory shows consistent identification of valuable points across all removal budgets. This suggests that the unlearning-based formulation captures a more nuanced understanding of data importance than gradient-based approaches.

Result on Adult presents an interesting edge case where all methods trigger near-immediate collapse after minimal removal (5-10%). This behavior likely reflects the dataset's small size and high feature dimensionality, creating a scenario where most training points become critical for maintaining the decision boundary. Here, Unlearning Shapley remains among the top performers, triggering collapse as early as any competing method.

The low-value removal results in the bottom row of Figure 1 provide equally important insights through the low-value removal task. Unlearning Shapley maintains comparable performance across datasets. On Breast Cancer, the method preserves high accuracy even after removing 60% of the lowest-valued points, compared to other methods that rapidly drop. We also notice the sudden performance drops on Adult, which may result from the instability of unlearning due to the simple feature of Adult.

This result of data removal demonstrates that Unlearning Shapley achieves superior value concentration. The method effectively separates truly influential examples from redundant ones, a critical property for practical applications like data pruning or active learning. The Adult dataset presents an anomaly where all methods perform similarly, likely due to the tabular nature and strong feature-label correlations that make most examples equally informative.

### 4.3 NOISY DATA DETECTION EXPERIMENT

We evaluate the ability of Unlearning Shapley to identify corrupted training examples through systematic label noise injection. Following the protocol established in OpenDataValJiang et al. (2023), we randomly flip 20% of training labels and assess whether valuation methods can distinguish these corrupted examples from clean ones. After computing data values for all training points, we apply k-means clustering ($k = 2$) to partition the data into two groups based on their values. The cluster with the lower mean value is designated as "detrimental" and predicted to contain the noisy examples. We then compute the F1-score between these predictions and the ground-truth corruption labels, where ground truth is used solely for evaluation and never during value computation.

The corrupted data detection results in Figure 2 demonstrate Unlearning Shapley's effectiveness in identifying label noise across diverse datasets. The method achieves strong early detection rates on MNIST, identifying over 80% of corrupted examples within the first 45% of examined data (first bin). This early detection capability is crucial for practical applications where manual inspection budgets

Table 1: **Noisy data detection.** We cluster the *computed* data values with $k$-means into two groups and label the cluster with the lower mean as *detrimental*; all points in this cluster are predicted as noisy. We then compute the F1-score between this prediction and the ground-truth corrupted set (the ground truth is *not* used when computing data values). Higher is better.

| Methods | F1-Score↑ | | | | |
|---|---|---|---|---|---|
| | FMNIST | CIFAR-10 | Adult | Breast Cancer | MNIST |
| Beta Shapley | **0.434** | 0.301 | 0.514 | 0.342 | 0.352 |
| Data Shapley | 0.365 | **0.363** | 0.514 | 0.336 | 0.356 |
| Unlearning Shapley | 0.362 | 0.334 | **0.523** | 0.566 | **0.478** |
| KNN Shapley | 0.412 | 0.335 | 0.523 | **0.611** | 0.393 |
| Influence Function | 0.319 | 0.330 | 0.523 | 0.473 | 0.353 |

Table 2: **Partial data valuation.** We simulate partial data valuation by unlearning the target dataset $\mathcal{D}_{tgt}$ and compute the Spearman's rank correlation coefficient between the exact Data Shapley value and Unlearning Shapley value, as well as between Unlearning Shapley value and the performance of the model retrained without it. ($p < 0.1$)

| Dataset | Spearman's rank Correlation | |
|---|---|---|
| | Performance of Retrained Model | Exact Shapley Value |
| **Adult** | -0.323 | 0.598 |
| **Breast Cancer** | -0.184 | 0.562 |
| **MNIST** | -0.257 | 0.275 |
| **CIFAR-10** | -0.652 | 0.280 |
| **FMNIST** | -0.188 | 0.164 |
| **Text Corpus** | -0.543 | 0.143 |

are limited. On MNIST, Adult, and Breast Cancer, Unlearning Shapley performs comparably to the best methods, maintaining competitive detection rates throughout the examination range.

Table 1 provides a comprehensive view through F1-scores, which balance precision and recall in corruption detection. Unlearning Shapley achieves the highest F1-score on Adult (0.523) and MNIST (0.478), demonstrating robust performance on both tabular and image data. The strong performance on MNIST (F1=0.478) is particularly noteworthy, as it suggests that the unlearning mechanism effectively captures the confusion introduced by label noise. When a model unlearns a mislabeled example, the performance often improves or remains stable, resulting in low or negative Shapley values.

However, the results also reveal dataset-specific variations. On Breast Cancer, KNNShapley achieves superior performance (F1=0.611), while BetaShapley leads on FashionMNIST (F1=0.434). This variation suggests that different valuation principles may be better suited to different data characteristics. Unlearning Shapley's competitive but not uniformly superior performance in noise detection indicates that while unlearning effectively identifies examples that harm model performance, the relationship between harmfulness and label corruption is complex and dataset-dependent.

The varying performance across datasets provides insights into how Unlearning Shapley operates. On datasets with clear class boundaries (e.g., MNIST, Adult), mislabeled points create obvious conflicts that unlearning readily identifies. However, on datasets with inherent class ambiguity, such as FashionMNIST, with similar clothing categories, even correctly labeled boundary cases might receive low values, complicating corruption detection. This observation suggests that Unlearning Shapley measures functional importance rather than labeling correctness.

## 4.4 PARTIAL DATA VALUATION

Partial data valuation represents a crucial yet underexplored scenario where practitioners need to evaluate a specific subset's contribution without access to the complete training data. This setting

naturally arises in federated learning, data marketplaces, and privacy-preserving ML deployments. We investigate whether Unlearning Shapley can provide reliable valuations using only the pre-trained model and target subset.

For Text Corpus, we use its three subsets (WikiText, FineWeb, Cosmopedia) to compute Unlearning Shapley values for each subset using Algorithm 2. For the remaining small-scale datasets, we randomly selected 10% of the data examples to evaluate through Algorithm 2 and the other 90% of the data examples as untouchable, retaining data every time, and we repeated this process 100 times. To establish ground truth, we also compute exact Shapley values. For Text Corpus, we follow the original Shapley value definition to retrain GPT2-XL on the combination of the three subsets and compute their exact Shapley value. For other datasets, we use the Shapley value obtained in full data valuation in the previous experiment. We then assess the correlation between Unlearning Shapley values and both exact Shapley values and retrained model performance.

The results in Table 2 reveal strong positive correlations between Unlearning Shapley and exact Shapley values across all datasets, with coefficients ranging from 0.143 (Text Corpus) to 0.598 (Adult). These significant positive correlations ($p < 0.1$) validate that unlearning-based approximation captures the essential contribution structure despite not requiring full data access. The variation in correlation strength reflects dataset complexity—simpler datasets with clear feature-label relationships (Adult, Breast Cancer) show stronger correlations, while complex, high-dimensional datasets (Text Corpus) present greater approximation challenges.

More importantly, the negative correlations between Unlearning Shapley values and retrained model performance provide empirical validation of the method's effectiveness. The strong negative correlation on CIFAR-10 (-0.652) indicates that subsets assigned high Unlearning Shapley values indeed cause substantial performance degradation when removed. This inverse relationship confirms that Unlearning Shapley correctly identifies influential data subsets without requiring expensive retraining experiments.

The Text Corpus results deserve special attention as they represent the most realistic large-scale scenario. Despite the modest correlation with exact Shapley values (0.143), the strong negative correlation with retrained performance (-0.543) suggests that Unlearning Shapley remains practically useful for identifying important data sources in LLM training. This finding is particularly valuable given the prohibitive cost of retraining large language models for exact valuation.

The consistency of negative correlations across diverse datasets, including tabular data (Adult: -0.323) to complex vision (CIFAR-10: -0.652) and language tasks (Text Corpus: -0.543), demonstrates that Unlearning Shapley provides a universally applicable solution for partial data valuation. This addresses a critical gap in existing methods, which require full data access and thus cannot operate in federated or privacy-preserving settings.

## 5  CONCLUSION

We propose Unlearning Shapley that leverages machine unlearning to extend Shapley-value-based data valuation to both full-data and partial-data settings. By removing the data from a pre-trained model, Unlearning Shapley enables data valuation without requiring full dataset access. We established its theoretical equivalence to classical Shapley values under mild assumptions, derived error bounds that separate unlearning approximation from Monte Carlo variance, and proposed practical algorithms for different deployment scenarios.

Empirical experiments show that Unlearning Shapley performs comparably with existing methods, while uniquely succeeding in privacy-sensitive partial-data scenarios. It correlates positively with exact Shapley values and negatively with model performance after data removal, supporting its reliability when only limited data access is available.

However, we also acknowledge several limitations. The performance depends on the fidelity of the unlearning process, which can vary across architectures and hyperparameters, and the experiments are confined to classification tasks, while extending the framework to generative modeling and continual learning remains an open direction.

In summary, this work broadens the scope of machine unlearning from privacy and compliance to principled data valuation. Unlearning Shapley provides a foundation for privacy-compliant data markets and federated learning ecosystems.

ETHICS STATEMENT

All experiments were conducted on publicly available benchmark datasets, without the use of human subjects or sensitive personal information. The proposed method is intended to advance fair and privacy-preserving data valuation, and we have transparently reported limitations to mitigate potential misuse.

REPRODUCIBILITY STATEMENT

We have made every effort to ensure reproducibility of our results. All datasets used in this study are publicly available, and we provide detailed descriptions of model architectures, training hyperparameters, and evaluation protocols in the main text and Appendix B. Algorithms for both full-data and partial-data valuation are presented in Algorithms 1 and 2, respectively.

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

Table 3: Notation used in the theoretical analysis.

| Symbol | Meaning |
|---|---|
| $N = \{d_i\}_{i=1}^n$ | Full training dataset with $n$ data points |
| $S \subseteq N$ | Subset of training data |
| $v(S)$ | Expected test performance of a model trained from scratch on $S$ |
| $u(S)$ | Test performance after unlearning $S$ from $\mathcal{M}_{\text{full}}$ |
| $\mathcal{M}_{\text{full}}$ | Pre-trained model on the full dataset $N$ |
| $\widehat{\mathcal{M}}_S$ | Model obtained after unlearning subset $S$ from $\mathcal{M}_{\text{full}}$ |
| $S_\pi^i$ | Predecessors of element $i$ in permutation $\pi$ |
| $T_\pi^i$ | Successors of element $i$, i.e., $N \setminus (S_\pi^i \cup \{i\})$ |
| $\phi_i$ | Shapley value of data point $i$ |
| $\Delta_i^\star(S)$ | Ideal marginal: $v(S \cup \{i\}) - v(S)$ |
| $\hat{\Delta}_i(S)$ | Unlearning marginal: $u(S) - u(S \cup \{i\})$ |
| $\varepsilon_u$ | Maximum deviation between $v(S)$ and $u(N \setminus S)$ |
| $\delta_u$ | Probability parameter for the unlearning accuracy bound |
| $\delta_{eval}$ | failure probability of evaluation/test-set concentration bound |
| $M$ | Number of sampled permutations for Monte Carlo estimation |
| $C_{\text{train}}$ | Cost of training a model from scratch |
| $C_{\text{unlearn}}$ | Cost of performing one unlearning update |
| $C_{\text{eval}}$ | Cost of evaluating a model on the test set $\mathcal{T}$ |

## A  THEORETICAL ANALYSIS OF UNLEARNING SHAPLEY

Let $N = \{d_i\}_{i=1}^n$ be the training set. For a subset $S \subseteq N$, define

$$v(S) := \mathbb{E}[\text{Perf}(\mathcal{M}_S)] \qquad \text{and} \qquad u(S) := \text{Perf}(\widehat{\mathcal{M}}_S),$$

where $v(S)$ denotes the expected test performance of a model trained from scratch on $S$, and $u(S)$ denotes the test performance of the full model $\mathcal{M}_{\text{full}}$ after unlearning $S$. For a permutation $\pi$ over $N$, let $S_\pi^i = \{j \in N : \pi(j) < \pi(i)\}$ be the predecessors of element $i$ and $T_\pi^i = N \setminus (S_\pi^i \cup \{i\})$ the successors.

The central modeling principle is a complement–unlearning approximation: training on $S$ is approximated by unlearning its complement from a well-trained full model. Formally, we assume that for all $S \subseteq N$,

$$\left|v(S) - u(N \setminus S)\right| \leq \varepsilon_u \quad \text{with probability at least } 1 - \delta_u, \tag{6}$$

and that unlearning is sequentially composable, i.e., moving from $\widehat{\mathcal{M}}_S$ to $\widehat{\mathcal{M}}_{S \cup \{i\}}$ does not require reinitialization from $\mathcal{M}_{\text{full}}$ beyond negligible overhead.

Under assumption equation 6, the standard Shapley marginal contribution can be rewritten:

$$v(S_\pi^i \cup \{i\}) - v(S_\pi^i) \approx u(N \setminus (S_\pi^i \cup \{i\})) - u(N \setminus S_\pi^i)$$
$$= u(T_\pi^i) - u(T_\pi^i \cup \{i\}), \tag{7}$$

where $T_\pi^i = N \setminus (S_\pi^i \cup \{i\})$ are the successors of $i$ in permutation $\pi$.

The key insight is that averaging over all permutations with successor sets is equivalent to averaging with predecessor sets. Specifically, for any permutation $\pi$, consider its reverse permutation $\pi'$ where the ordering is reversed. The successors of $i$ in $\pi$ become related to the predecessors of $i$ in $\pi'$. Since this reversal is a bijection on the set of all permutations $\Pi$:

$$\mathbb{E}_{\pi \in \Pi}[u(T_\pi^i) - u(T_\pi^i \cup \{i\})] = \mathbb{E}_{\pi \in \Pi}[u(S_\pi^i) - u(S_\pi^i \cup \{i\})]. \tag{8}$$

This yields the unlearning form of the Shapley value:

$$\phi_i \approx \mathbb{E}_\pi \left[u(S_\pi^i) - u(S_\pi^i \cup \{i\})\right], \tag{9}$$

which represents the expected drop in performance when additionally unlearning $i$ after having already unlearned $S_\pi^i$. Equation 9 leads directly to a Monte Carlo estimator that traverses sampled permutations and accumulates $u(S) - u(S \cup \{i\})$ along the unlearning path.

Table 4: The Elapsed time of model training and unlearning on a given amount of data.

| Dataset | Data Scale | Elapsed time | |
| --- | --- | --- | --- |
| | | Unlearn | Train |
| **FMNIST (MLP)** | 60,000 | $0.992 \pm 0.155$s | $20.035 \pm 0.584$s |
| **CIFAR-10 (ResNet-18)** | 50,000 | $23.333 \pm 0.911$s | $3.132 \pm 0.062$m |
| **Text Corpus Subsets(GPT2-XL)** | 1,000,000 | $5.435 \pm 0.067$m | $17.669 \pm 0.584$h |

For partial valuation, when only a target subset $\mathcal{D}_{\text{tgt}}$ and the full model are available, data valuation reduces to a two-player game with players $\mathcal{D}_{\text{tgt}}$ and $\mathcal{D}_{\text{retain}} = N \setminus \mathcal{D}_{\text{tgt}}$. The classical two-player Shapley formula gives

$$\phi_{\text{tgt}} = \tfrac{1}{2}[v(\mathcal{D}_{\text{tgt}}) - v(\emptyset)] + \tfrac{1}{2}[v(\mathcal{D}_{\text{tgt}} \cup \mathcal{D}_{\text{retain}}) - v(\mathcal{D}_{\text{retain}})].$$

Note that $v(\mathcal{D}_{\text{tgt}} \cup \mathcal{D}_{\text{retain}}) = v(N)$ and applying equation 6:

$$v(\emptyset) \approx u(N), \quad v(\mathcal{D}_{\text{tgt}}) \approx u(\mathcal{D}_{\text{retain}}),$$
$$v(\mathcal{D}_{\text{retain}}) \approx u(\mathcal{D}_{\text{tgt}}), \quad v(N) \approx u(\emptyset). \tag{10}$$

This yields the implementable closed form:

$$\phi_{\text{tgt}} \approx \tfrac{1}{2}\big[u(\emptyset) - u(\mathcal{D}_{\text{tgt}}) + u(\mathcal{D}_{\text{retain}}) - u(N)\big], \tag{11}$$

where $u(\emptyset) = \text{Perf}(\mathcal{M}_{\text{full}})$ and $u(N)$ corresponds to unlearning all data (practically approximated by the performance of a randomly initialized model). If $\mathcal{D}_{\text{retain}}$ is inaccessible, we estimate $u(\mathcal{D}_{\text{retain}}) \approx v(\mathcal{D}_{\text{tgt}})$ by training a model from scratch on $\mathcal{D}_{\text{tgt}}$, which is consistent with equation 6.

**Error Analysis.** To quantify deviation from retraining-based Shapley, let $\Delta_i^{\star}(S) = v(S \cup \{i\}) - v(S)$ and $\hat{\Delta}_i(S) = u(S) - u(S \cup \{i\})$. Assumption equation 6 implies

$$\big|\hat{\Delta}_i(S) - \Delta_i^{\star}(S)\big| \leq 2\varepsilon_u \quad \text{with probability at least } 1 - 2\delta_u,$$

which captures the unlearning approximation bias. Even for exact $v(\cdot)$, optimization randomness induces variability in $\Delta_i^{\star}(S)$; under standard Lipschitz and sub-Gaussian conditions on model outputs, this contributes an additive term $B_{\text{rand}}(S, i)$. If $\hat{\phi}_i = \frac{1}{M} \sum_{m=1}^{M} \hat{\Delta}_i(\pi_m)$ averages $M$ i.i.d. permutation marginals, Hoeffding's inequality gives a Monte Carlo error at most $B\sqrt{\frac{1}{2M} \ln \frac{2}{\eta}}$ with probability $1 - \eta$, where $B$ bounds each marginal.

**Unlearning Implementation.** The unlearning utility $u(\cdot)$ is instantiated by an approximate unlearning objective that maximizes loss on the deletion set while stabilizing parameters and outputs:

$$\mathcal{L}_{\text{unlearn}} = -\mathcal{L}_{\text{CE}}(\mathcal{D}_{\text{unlearn}}) + \lambda_1 \|\theta - \theta_{\text{full}}\|^2 + \lambda_2 \, D_{\text{KL}}\big(f_\theta(x) \,\|\, f_{\theta_{\text{full}}}(x)\big), \quad x \sim \mathcal{T}.$$

Under $L$-smoothness and a local Polyak–Łojasiewicz condition with constant $\mu > 0$, gradient descent with step size $\eta \in (0, 2/L)$ converges linearly to a task-dependent minimizer $\theta_S^{\star}$. The parameter and KL regularizers bound the parameter shift $\|\theta_S^{\star} - \theta_{\text{full}}\|$ and output shift respectively, yielding a bound on $|u^{\star}(S) - v(S)|$ and therefore on $\varepsilon_u$.

Combining these components, the total deviation of the Monte Carlo estimator from the retraining-based Shapley value satisfies, with probability at least $1 - \eta - 2\delta_u - \delta_{\text{eval}}$,

$$\big|\hat{\phi}_i - \phi_i\big| \leq 2\varepsilon_u + B_{\text{rand}} + B\sqrt{\frac{1}{2M} \ln \frac{2}{\eta}}.$$

# B TRAINING DETAILS

We set the parameters in training as follows. For MLP on FMNIST, we set the batch size to 32, epochs to 10, and learning rate to 0.0003 for an AdamW optimizer with betas (0.9, 0.999), eps set to 1e-8, and weight decay to 0.01. The hidden dimension of MLP is set to 512. For ResNet-18 on CIFAR-10, the learning rate is set to 0.0003, and other parameters are the same as MLP, and the

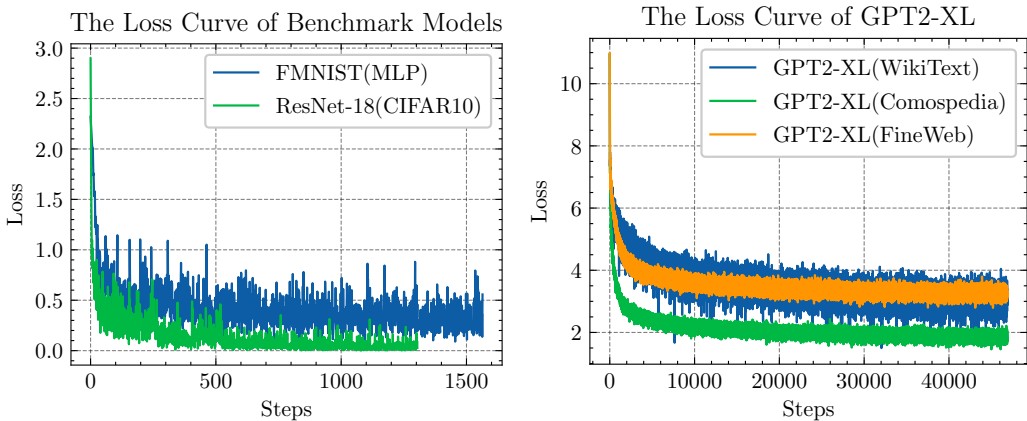

Figure 3: The loss curve of models on benchmark datasets and text Corpus.

Table 5: Evaluation of Unlearning on different datasets and models (NM-Not Meaningful)

| Dataset(Model) | Model Type | Metrics | | | |
|---|---|---|---|---|---|
| | | LKD | LMSE | KR | SPC |
| **CIFAR-10(ResNet-18)** | Unlearn | 0.825 | 0.077 | $\approx 0.000$ | 0.665 |
| | Random | 2.582 | 0.090 | NM | NM |
| **FMNIST(MLP)** | Unlearn | 0.781 | 0.018 | $\approx 0.000$ | -0.104 |
| | Random | 2.300 | 0.072 | NM | NM |
| **Text Corpus(GPT2-XL)** | Unlearn | 8.124 | $7.440 \times 10^{-6}$ | 0.377 | 0.812 |
| | Random | 10.96 | $4.120 \times 10^{-6}$ | NM | NM |

ResNet-18 is initialized from huggingface[1]. For the text corpus, the text length is cropped to 768. We initialize GPT2-XL[2]. The training epoch is set to 5, and the learning rate is set to 3e-4. For other datasets (Adult, MNIST, Breast Cancer), we train MLP and LeNet using Adam with a learning rate of 0.0003, batch size of 16, and train for 10 epochs. The models are trained on RTX5000.

For unlearning parameters, the weights $\lambda_1$ and $\lambda_2$ are set to 1.0. The loss curve is shown in Figure 3. The learning rate is set to 0.0003 for Adult, 0.0001 for Breast Cancer, 0.00002 for CIFAR-10, MNIST, FMNIST, and Text Corpus. The unlearning step is set to 250 for Text Corpus, and 125 for other datasets. The number of Monte-Carlo samples is set to 1000 for Shapley value approximation for Data Shapley, BetaShapley, and Unlearning Shapley.

## C  UNLEARNING EXPERIMENT

The error is hard to avoid for approximate unlearning. We conducted experiments on the unlearning process to evaluate whether it can completely unlearn $\mathcal{D}_{forget}$ with less impact on the remaining data $\mathcal{D}_{retain}$. Besides, we evaluate the efficiency of unlearning compared to training.

We separately trained models $\mathcal{M}_{retrain}^i$ on $\mathcal{D}_{retain}^i$, which excludes $\mathcal{D}_i$ from $\mathcal{D}_{full}$. Besides, we used $\mathcal{M}_{full}$ in Section 4.1 to unlearn each subsets $\mathcal{D}_i$ to obtain $\mathcal{M}_{unlearn}^i$ for comparison. We also introduce an initialized model $M_{random}$. For the benchmark datasets, we have $\mathcal{D}_{full} = \bigcup_{i=1}^{C} \mathcal{D}_i$, where $C$ is the number of classes. Then, we divide it into subsets by their class $\mathcal{D}_i = \{(x,y) \in \mathcal{D}_{full} \mid y = i\}$. As for Text Corpus, we merely regard Wikitext, Fineweb, and Cosmopedia as three subsets.

---

[1] https://huggingface.co/microsoft/resnet-18
[2] https://huggingface.co/openai-community/gpt2

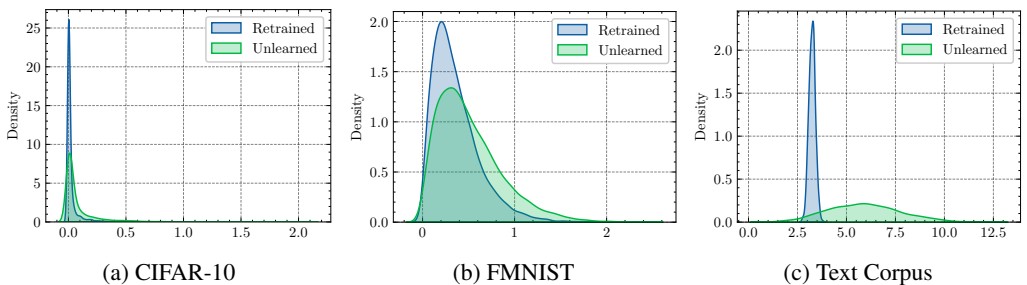

(a) CIFAR-10 (b) FMNIST (c) Text Corpus

Figure 4: The kernel density estimation of models' losses on $\mathcal{D}_{retain}$.

We evaluate how well the model can unlearn given data, and how big the bias is between the $\mathcal{M}_{unlearn}$ and $\mathcal{M}_{retrain}$. The metrics are as follows:

**Logit Kullback-Leibler Divergence (LKD)**. LKD measures the decision bias of the trained and unlearned model on the remaining data, where high LKD means greater damage the unlearning process has on the remaining data. Given a data example, logits $\{0.60, 0.30, 0.10\}$ and $\{0.80, 0.03, 0.17\}$ both get the correct result, i.e., same metrics, but we can hardly tell they are nearly the same model for the difference in their output distribution. Therefore, we compute the KLD between the output logit of the $\mathcal{M}_{unlearn}$ and $\mathcal{M}_{retrain}$ on the remaining set $\mathcal{D}_{retain}$.

**Logit Mean Square Error (LMSE)**. Besides LKD, we also compute the MSE error between the output logit of the $\mathcal{M}_{unlearn}$ and $\mathcal{M}_{retrain}$ on $\mathcal{D}_{retain}$ to measure the impact of unlearning on the remaining set.

**Knowledge Retention (KR)**. KR measures how complete the unlearning is on the unlearned data as follows:

$$\frac{Perf(\mathcal{M}_{unlearn}, \mathcal{D}_{forget}) - Perf(\mathcal{M}_{random}, \mathcal{D}_{forget})}{Perf(\mathcal{M}_{full}, \mathcal{D}_{forget}) - Perf(\mathcal{M}_{random}, \mathcal{D}_{forget})} \tag{12}$$

KR ranges from 0 to 1, where a lower KR stands for a lower similarity of $\mathcal{M}_{unlearn}$ and $\mathcal{M}_{full}$ on $\mathcal{D}_{forget}$, i.e., more complete unlearning. We use accuracy for classification tasks and negative perplexity for the text corpus in the experiment.

**Spearman's rank correlation coefficient (SPC)**. SPC means the correlation between the performance of the unlearned model $\mathcal{M}_{unlearn}$ and the model retrained on the remaining data; they should have similar performance if the unlearning is accurate. Ranging from -1.0 to 1.0, a higher SPC stands for more accurate unlearning.

We computed the above metrics on each $\mathcal{D}_i$ and averaged them as the results in Table 5. $\mathcal{M}_{unlearn}$ shows lower LKD and LMSE compared with $\mathcal{M}_{random}$ on $\mathcal{D}_{retain}$. We can see that $\mathcal{M}_{unlearn}$ shows approximate performance and decision to $\mathcal{M}_{retrain}$ on benchmark datasets. Also, $\mathcal{M}_{unlearn}$ leaks no knowledge on the $\mathcal{D}_{unlearn}$ when compared with $\mathcal{M}_{full}$. However, we also noticed $\mathcal{M}_{unlearn}$ got large LKD and KR on the text corpus due to the huge vocabulary size of LLMs, which for GPT2-XL is 50,257 Radford et al. (2019). Therefore, even slight differences in logits can result in huge LKD. Unlike the image classification task, there is more shared knowledge in the text corpus; this makes $\mathcal{M}_{unlearn}$ still retain knowledge on $\mathcal{D}_{forget}$ because of the overlapping information in $\mathcal{D}_{forget}$ and $\mathcal{D}_{retain}$, which contribute to the high KR in the text corpus.

We visualized the difference of $\mathcal{M}_{retrain}$ and $\mathcal{M}_{unlearn}$ in their loss on the $\mathcal{D}_{retain}$ by kernel density estimation (KDE), which is depicted in 4. We find that the loss of $\mathcal{M}_{unlearn}$ and $\mathcal{M}_{retrain}$ on $\mathcal{D}_{retain}$ follows the similar distribution on CIFAR-10 and FMNIST. The loss gap between the two models on the text corpus may also be attributed to the overlapping of the text corpus, i.e., unlearning one subset can easily cause damage to other data by losing the shared knowledge. We notice the high SPC in CIFAR-10 and Text Corpus, proving the accuracy of unlearning compared with unlearning FMNIST, which may be attributed to the simple feature space in FMNIST.

We also compared the elapsed time of the unlearning and training process on subsets (Appendix C.1. The gap between the elapsed time becomes larger with the increase of the model and data scales. We

find for GPT2-XL on a subset with 1,000,000 points, unlearning can reach even 195 times faster than training.

### C.1 COMPUTATIONAL EXPERIMENT ON UNLEARNING

Considering that unlearning is more efficient than model training, we experimented to evaluate the elapsed time of unlearning a given amount of data. The results are shown in Table 4; because several steps are enough for unlearning, unlearning is without doubt faster than training to convergence. Under the setting in Appendix B, the gap of the elapsed time between training and unlearning becomes larger, even reaching nearly 195 times for GPT2-XL on Text Corpus.

## THE USE OF LARGE LANGUAGE MODELS

In preparing this manuscript, we utilized large language models (LLMs) to assist with language editing and refinement. Specifically, LLMs were utilized to enhance the clarity, grammar, and readability of the text. All technical content, theoretical analysis, experimental design, and interpretation of results were conceived, implemented, and verified by the authors. The role of the LLMs was limited to surface-level writing assistance, and the authors are fully responsible for the correctness and originality of the scientific contributions presented in this work.

