# OpenReview forum: "Unlearning Shapley: Data Valuation through Machine Unlearning"
_ICLR.cc/2026/Conference — ICLR 2026 Conference Withdrawn Submission_

### Official Review · Reviewer_24EF · 2025-10-24

**Soundness:** 2
**Presentation:** 3
**Contribution:** 2
**Rating:** 4
**Confidence:** 4

**Summary:**

This paper proposes Unlearning Shapley, which estimates data value by unlearning subsets from a pretrained model instead of retraining on all coalitions. The paper shows the connection between Unlearning Shapley and classical Shapley through theoretical analysis, and provides bias and error bounds for their estimator.

**Strengths:**

1. The Unlearning Shapley is a clear reduction from retraining.

2. The writing is clear and easy to follow.

**Weaknesses:**

1. A key concern for the reviewer is that this work looks just a simple application of unlearning method, with only limited contributions. And it's performance is highly relied on the selected unlearning methods.

2. For the approximate unlearning method used in the paper, the equation above the experiment setion, authors use a test set to replace the retain set. Is it reasonable to use the test dataset in the unlearning training? There are many unlearning methods that do not use the remaining data [R1, R2].

[R1]. Tarun, Ayush K., et al. "Fast yet effective machine unlearning." IEEE Transactions on Neural Networks and Learning Systems 35.9 (2023): 13046-13055.

[R2]. Panda, Subhodip, and Shashwat Sourav. "Partially Blinded Unlearning: Class Unlearning for Deep Networks from Bayesian Perspective." Proceedings of the AAAI Conference on Artificial Intelligence. Vol. 39. No. 6. 2025.

3. The theoretical bound focuses on closeness between unlearning and retraining utility, but does not provide a guarantee that the model no longer used information from the deleted data.

4. In the experimental section, it has not introduced the purposes of these experiments. In Table 1, it seems that Unlearning Shapley only achieves a good performance on Adult and MNIST. And all these datasets are toy datasets, we should evaluate the method in a more complex dataset.

**Questions:**

See above weaknesses.

**Details Of Ethics Concerns:**

No ethics concerns.

---

### Official Review · Reviewer_ZUSd · 2025-10-29

**Soundness:** 3
**Presentation:** 3
**Contribution:** 2
**Rating:** 4
**Confidence:** 4

**Summary:**

This paper introduces Unlearning Shapley, a framework that approximates Shapley values by replacing expensive retraining with machine unlearning. The primary contribution is an efficient data valuation method, especially for partial-data scenarios where existing methods are inapplicable.

**Strengths:**

The paper's primary strength is its novel idea of connecting machine unlearning with Shapley value estimation. This approach tackles the significant and practical problem of partial data valuation.

**Weaknesses:**

1. The paper's core approximation (Eq. 3) lacks validation (e.g., quantifying $\varepsilon_{u}$), and the impact of different unlearning algorithms or configurations is not studied.
2,  The claimed superiority of the method is not supported by the majority of the results in Figure 1 and Table 1. The explanation for its unique failure on the Adult dataset (lines 356-358) is particularly unconvincing. And the paper repeatedly emphasizes its applicability to large models, yet the experiments conducted on them are very limited.

**Questions:**

1. Beyond approximation error, does the imperfect unlearning process introduce a systematic bias into the Shapley values?
2. Could you provide a deeper analysis for the method's inconsistent performance where it often fails to outperform baselines, and explain the root cause of its unique failure on the Adult dataset?
3. The paper claims superior efficiency over retraining-based methods. Could you provide a full, end-to-end time comparison between the Unlearning Shapley algorithm and a baseline?

---

### Official Review · Reviewer_LPLG · 2025-10-30

**Soundness:** 1
**Presentation:** 2
**Contribution:** 2
**Rating:** 2
**Confidence:** 4

**Summary:**

Overall, this is an interesting study that tackles an important and practical problem for Shapley value-based XAI tools by incorporating ideas from machine unlearning. The novelty is sufficient; however, I have several questions regarding the current experimental design and results. It would be helpful if the authors could justify these and, ideally, provide more detailed experiments to strengthen the paper's contributions. My detailed questions are listed below.

**Strengths:**

The research gap this paper addresses is relevant and has the potential for significant impact in the field. The methods are clearly written and theoretically justified. The datasets used for evaluation are comprehensive, covering both structured and unstructured data. The overall writing is clear and straightforward.

**Weaknesses:**

The current results presented lack sufficient evidence to justify that the proposed method is The current results lack sufficient evidence to justify that the proposed method has significant advantages over existing ones. For instance, in Table 1, the F1 score of the proposed method is outperformed by baselines in several datasets.

Furthermore, no confidence intervals or measures of variance (e.g., from repeating experiments 10 times) are reported in Table 1. This lack of statistical rigor makes the authors' own conclusion—that "Unlearning Shapley’s competitive but not uniformly superior performance in noise detection indicates that while unlearning effectively identifies examples that harm model performance, the relationship between harmfulness and label corruption is complex and dataset-dependent."—sound weak and unsubstantiated.

In Table 2, the Spearman correlation is quite low (all < 0.6), suggesting the approximation is not very reliable. More importantly, the results for all baseline methods are omitted. This makes it impossible to validate the paper's central claim: that it uniquely succeeds in the partial-data setting where other methods "fail."

**Questions:**

1. Can the authors please add results from multiple runs (e.g., mean and standard deviation) for Table 1 to demonstrate that their method's performance is statistically significant and not due to chance?

2. Regarding Table 2, can the authors justify why the baseline methods are omitted? If it is not impossible for them to run, their results must be included for a fair comparison. If it is impossible, this should be explicitly justified and, if possible, demonstrated.

---

### Official Review · Reviewer_vVYA · 2025-10-31

**Soundness:** 2
**Presentation:** 2
**Contribution:** 2
**Rating:** 2
**Confidence:** 4

**Summary:**

This paper introduces Unlearning Shapley, a framework that uses machine unlearning to enable valuation when only a target subset of data is available and may be more efficient than computing the Shapley value. The key idea is to approximate the utility of a data subset by unlearning a complement set from the fully trained model instead of retraining from scratch.

**Strengths:**

Unlearning Shapley is original in the use of unlearning for data valuation and may be important for efficient data valuation of large models, such as LLMs when only a small subset is removed. The paper also identifies a reasonable use case of doing data valuation when the training data is not fully available.

**Weaknesses:**

1. The contributions of the paper may be limited. The key idea is to approximate the utility of a data subset by unlearning a complement set from the fully trained model instead of retraining from scratch. In other words, it suggests a known approximation for the utility function instead of a more efficient way to approximate the Shapley value or a new data value that is an alternative to the Shapley value.
    * The approximation quality depends on the unlearning approximation error $\epsilon_u$ but this is unknown and not analysed in the experiments. The assumption (Eq 6) should appear in the main paper and be justified.
2. There is a lack of comparison with other unlearning methods and data valuation approximation methods.
    * The paper suggests a new approximate unlearning algorithm in Sec 3 but there is no theoretical or empirical guarantee that it approximates retraining well and is better than other methods.
    * The related work section does not discuss other efficient approximation techniques to approximate the utility function e.g., using a single epoch in Data Shapley.
3. The empirical results are not strong.
    * In Fig 1 and 2,  Unlearning Shapley does not seem to outperform other methods on most experiments.
    * More comprehensive benchmark experiments can be performed using OpenDataVal.
    * The experiments do not analyse the quality of the utility approximation directly.
    * In Table 2, the correlation with the exact Shapley value is only moderately high (above .5) for 2 datasets. A perfect approximation method should have a correlation of 1 for exact Shapley value but there is no ideal value for correlation with the performance of the retrained model so it may be less meaningful.

**Questions:**

1. Explain the mild assumptions behind Equation (5). Specifically, also explain why the unlearning approximation error $\epsilon_u$ is expected to be low for unlearning algorithms, including the new one proposed.

Minor comments:
* For equation (3), it may be clearer to use $T$ instead of $S$.
* The citation should be in (author, year) instead of author (year)

---

### Note · Authors · 2025-11-26

I have read and agree with the venue's withdrawal policy on behalf of myself and my co-authors.